# Early Clinical Remission Is a Predictor of Long-Term Remission with the Use of Vedolizumab for Ulcerative Colitis

**DOI:** 10.3390/biomedicines10102526

**Published:** 2022-10-09

**Authors:** Keiichi Haga, Tomoyoshi Shibuya, Taro Osada, Shunsuke Sato, Yuka Fukuo, Osamu Kobayashi, Toshio Yamada, Daisuke Asaoka, Kentaro Ito, Kei Nomura, Mayuko Haraikawa, Osamu Nomura, Hirofumi Fukushima, Takashi Murakami, Dai Ishikawa, Mariko Hojo, Akihito Nagahara

**Affiliations:** 1Department of Gastroenterology, Juntendo University School of Medicine, 2-2-1 Hongo, Bunkyoku, Tokyo 113-8421, Japan; 2Department of Gastroenterology, Juntendo University Urayasu Hospital, 2-1-1 Tomioka, Urayasu-shi 279-0021, Japan; 3Department of Gastroenterology, Juntendo University Shizuoka Hospital, 1129 Nagaoka, Izunokuni-shi 410-2295, Japan; 4Department of Gastroenterology, Juntendo University Nerima Hospital, 3-1-10 Takanodai, Nerima-ku, Tokyo 177-8521, Japan; 5Department of Gastroenterology, Koto Hospital, 6-8-5 Oshima, Koto-ku, Tokyo 136-0072, Japan; 6Department of Gastroenterology, Tokyo Rinkai Hospital, 1-4-2 Rinkai-cho, Edogawa-ku, Tokyo 134-0086, Japan; 7Department of Gastroenterology, Juntendo Tokyo Koto Geriatric Medical Center, 3-3-20 Shinsuna, Koto-ku, Tokyo 136-0075, Japan

**Keywords:** inflammatory bowel disease, ulcerative colitis, Vedolizumab

## Abstract

Vedolizumab (VDZ) is an α4β7 integrin-antibody used to manage refractory ulcerative colitis (UC). This retrospective multicenter study aimed to identify predictors of efficacy or the time points when evaluation of VDZ therapy for UC would be most useful. We compiled data on 87 patients with moderate to severe active UC that was treated with VDZ. Overall clinical remission (CR) rates at 6 weeks and 52 weeks after VDZ administration were 44.4% (bio-naïve 44.2%, bio-failure 44.8%) and 52.8% (bio-naïve 53.5%, bio-failure 51.7%) respectively. Also, 83.3% (bio-naïve 81.3%, bio-failure 85.7%) of patients achieved mucosal healing at week 52. Among patients with a CR at week 52, 73.3% had a CR at week 6. In contrast, of patients who discontinued VDZ, 82.4% had not reached a CR at week 6. Our study demonstrated that VDZ was effective in a large percentage of UC patients, with a high mucosal healing rate even after prior biological exposures. This suggests that VDZ can be a treatment option even in bio-failure cases. Additionally, it was considered that early CR can predict long-term remission and that week 6 can be a helpful evaluation point for treatment decisions when using VDZ for UC.

## 1. Introduction

Ulcerative colitis (UC) is a chronic relapsing and remitting inflammatory disorder of the large intestine characterized by bloody diarrhea and abdominal pain [1]. In recent years, treatment of UC has made remarkable progress. Various therapeutic agents with different mechanisms have been developed, and biological drugs have become important in the management of UC [2]. While various molecular targeting medications, including anti-tumor necrosis factor-α (TNF-α) antibody, have greatly improved therapeutic strategies for UC, these drugs may be associated with other problems, such as serious infections or loss of response [3]. 

Vedolizumab (VDZ) is a humanized monoclonal antibody specifically targeting α4β7 integrin. VDZ inhibits binding between α4β7 integrin and mucosal vascular addressin cell adhesion molecule-1 (MAdCAM-1) expressed in the intestinal tract, which inhibits the migration of inflammatory cells into the gastrointestinal tract [4,5,6]. VDZ does not block adhesion to vascular cell adhesion molecule-1 (VCAM-1) expressed in vascular endothelial cells throughout the body. Although it is useful in the treatment of inflammatory bowel disease (IBD) by inhibiting intestinal lymphocyte infiltration, it has little effect on organs other than the intestinal tract, and it is said to be safer than conventional anti-TNF-α antibody formulations [7]. The GEMINI-1 trial is a randomized, placebo-controlled large-scale clinical study evaluating the efficacy of VDZ for treatment of active UC. The clinical response rate 6 weeks after the administration of VDZ was 47.1%, and the 52-week clinical remission (CR) rate was 41.8%, both of which were significantly higher compared to that of the placebo [8]. There was no significant difference in adverse events between the VDZ group and the placebo group. VDZ has been reported to be useful for the induction and maintenance of remission in UC and contributes to the improvement of QOL [9]. The number of reports on the efficacy and safety of long-term VDZ administration have been increasing. Loftus et al. reported that in UC patients who achieved a clinical response with VDZ administration, the maintenance of remission was 88% at 104 weeks and 96% at 152 weeks [10]. Motoya et al. conducted a randomized, placebo-controlled phase 3 study that included 292 patients with active UC from Japan. The clinical response rate at week 10 was 39.6%, and the CR rate at week 60 was 56.1% in patients treated with VDZ, both of which were significantly higher than in the placebo group [11]. VDZ has been suggested to have long-term effects in maintaining remissions rather than only short-term effects [12]. 

In recent years, the goal of UC treatment has shifted to endoscopic remission and mucosal healing [13,14]. However, blood sampling data and clinical symptoms sometimes deviate from endoscopic findings, and endoscopic inflammation may be observed even in patients who have achieved a CR [15,16]. Fecal calprotectin and leucine-rich alpha-2 glycoprotein (LRG) are used as predictors of intestinal inflammation. Fecal calprotectin levels of 250 μg/g or less indicate a high probability of endoscopic remission. It was reported that fecal calprotectin values rise three months before relapse; thus, measuring it periodically makes it possible to predict the risk of relapse [17,18]. In addition, LRG was reported to be more accurate in assessing endoscopic severity than conventional C-reactive protein (CRP) [19]. However, no biomarkers or clinical characteristics have been established that can predict long-term prognosis or endoscopic remission as an aid in deciding to change or intensify treatment for UC. When administering VDZ, it is important to assess the improvement in clinical symptoms and to appropriately determine whether maintenance treatment can be continued [20]. There are few reports on the examination of predictors of therapeutic efficacy or on factors related to evaluations of treatment decisions when using VDZ. The aim of this study was to identify predictors of efficacy or time points most useful for the evaluation of VDZ therapy for UC. 

## 2. Methods

### 2.1. Study Design

This is a retrospective multi-center study. Data were compiled from 87 patients with moderate to severe active UC who were treated with VDZ between November 2018 and May 2022 in 6 centers across Japan. Patients who were followed for at least 52 weeks after VDZ induction were included in this study. Their medical records were reviewed retrospectively for information on diagnosis, clinical course, treatment, and laboratory parameters, including CRP, hemoglobin, and albumin. Patient data were registered into an electronic database after a deidentification process. The protocol for this retrospective investigation was reviewed and approved by the Juntendo University Hospital Ethics Committee (IRB no. H20-0013). This study was conducted in accordance with standards for Good Clinical Practice and adhered to the principles of the Declaration of Helsinki. Local and regional regulatory requirements were adhered to in each study center. Institutional review board approval was obtained from all study centers prior to study initiation. 

### 2.2. Patients

UC diagnosis was based on established standardized criteria by prior clinical assessment, radiology, endoscopy, and histology [21].

### 2.3. Treatment

The standard intravenous induction dose (300 mg) of VDZ was administered at weeks 0, 2, and 6, followed by maintenance therapy of an intravenous infusion every 8 weeks. The dosage for maintenance therapy was 300 mg, the same as the induction dosage. No patient required a shortened treatment interval or dose adjustment.

### 2.4. Definition of Response

At 6 and 52 weeks after initiating VDZ, clinical outcomes were assessed. Clinical disease activity was determined using the Lichtiger clinical activity index (CAI) [22], which elicits information on the following: number of daily bowel movements, abdominal pain and tenderness, use of antidiarrheals, blood in stools, general well-being, fecal incontinence, and nocturnal diarrhea. A higher score indicates a more severe disease (score range 0–21). CAI ≥ 10, 7–9, and ≤ 6 were defined as severe, moderate, and mild, respectively [23,24]. A decrease in the CAI of 3 or more points indicated a clinical response, and a score of 3 or fewer points a CR. Maintenance of efficacy was defined as no exacerbation of CAI and no need for intensifying treatments [25]. Endoscopic severity was determined using the Mayo endoscopic subscore (Mayo ES) classification (0, normal or inactive disease; 1, mild disease with erythema, decreased vascular pattern, mild friability; 2, moderate disease with marked erythema, absence of vascular patterns, friability, erosions; 3, severe disease with spontaneous bleeding, ulceration) [26]. Mucosal healing was defined as a Mayo ES of 0 or 1 [27].

### 2.5. Statistical Analysis

GraphPad Prism (version 6, GraphPad, La Jolla, CA, USA) was used to analyze all data. Between group differences were analyzed using Mann–Whitney’s U test. Relapse-free survival was assessed using the Kaplan–Meier method. *p* < 0.05 indicated significant differences. 

## 3. Results

### 3.1. Patient Characteristics

Table 1 shows the characteristics of the 87 study patients (47 males and 40 females, mean age 45.2 years (range 20–86 years), and a mean disease duration of 9.4 years (range 1–33 years)). Thirty-one and 56 patients were classified as having left-sided or extensive colitis, respectively. The mean Lichtiger CAI at baseline was 7.7 (range 0–17), and the median Mayo ES was 2. Thirty-four (39.1%) patients had a history of using biological drugs, including anti-TNF-α antibody. Eleven patients and 1 patient were treated with 2 and 3 biological drugs, respectively. Among those patients, 17 patients had a history of using infliximab (IFX), 16 used golimumab (GOM), and 12 used adalimumab (ADA). As a concomitant drug, 5-aminosalicylate (5-ASA), prednisolone (PSL), and Azathioprine (AZA) were used in 78 (89.7%), 36 (41.4%), and 33 (37.9%) patients, respectively. The mean doses of PSL and AZA were 6.3mg and 60.6mg, respectively. AZA and 5-ASA were combined in 28 patients, 27 patients were taking 5-ASA and PSL, and 4 patients were using a combination of 5-ASA, AZA and PSL. Seventy-two patients were treated with VDZ to induce remission and 15 patients who received drugs other than VDZ for induction therapy were provided VDZ for maintenance therapy. 

### 3.2. Treatment Efficacy

The overall CR rates in the 72 patients treated with VDZ for remission induction at week 6 and week 52 were 44.4% (bio-naïve 44.2%, bio-failure 44.8%) and 52.8% (bio-naïve 53.5%, bio-failure 51.7%) (Figure 1a,b), respectively. There was no significant difference between the bio-naïve and bio-failure groups at week 6 or week 52. CAI gradually decreased after the introduction of VDZ, and the average CAI at week 54 was 1.95 (Figure 1c). There was no significant difference in changes in CAI between the bio-naïve and bio-failure groups. Among the 30 patients who continued VDZ and underwent colonoscopy one year after VDZ initiation, 83.3% (bio-naïve 81.3%, bio-failure 85.7%) achieved mucosal healing. There was no significant difference between the bio-naïve and bio-failure groups (Figure 2a,b). The survival curve shows the cumulative non-relapse rate up to 100 weeks after VDZ administration, which is the percentage of cases that did not require surgery or change in medications due to exacerbation of symptoms. Again, no difference was observed between bio-naïve and bio-failure cases (Figure 3).

### 3.3. Treatment Efficacy in Bio-Failure Cases

Among the 34 patients previously treated with biological drugs, 22 patients (64.7%) used VDZ as a second bio and 11 patients (32.4%) as a third bio. Among the 34 patients, 28 cases were treated with VDZ to induce remission (second bio *n* = 18, third bio *n* = 10). CR rates in the 28 cases treated with VDZ as a second bio and third bio at week 6 were 61.1% and 20%, respectively, and the CR rates at week 52 were 61.1% and 40%, respectively (Figure 4a). The CR rate was significantly higher in the second bio group. At 52 weeks, 88.9% of the second bio and 80% of the third bio cases achieved mucosal healing, with no significant difference between the two groups (Figure 4b).

### 3.4. Analysis of VDZ Dropout Patients

At present, 33 patients have discontinued VDZ, and the average treatment period of these patients was 28 weeks. There was no significant difference in the treatment period between the bio-naïve and bio-failure groups (Figure 5). Among the 33 cases that discontinued VDZ, 18 switched to ustekinumab (UST), 7 to IFX, 3 to GOM, and 1 each to ADA and tofacitinib (TOF), which were selected by shared decision making. Three cases underwent surgery after discontinuation of VDZ. 

### 3.5. Early Clinical Remission Can Be a Predictor of Long-Term Remission

Among the 45 patients who had achieved a CR at week 52, 33 (73.3%) cases had showed a CR at week 6. In contrast, among the 33 patients who discontinued VDZ, 26 (78.8%) cases had not reached a CR at week 6. The cumulative non-relapse rate among patients with a CR at week 6 was significantly higher than that of cases who did not achieve a CR at that time (*p* < 0.0001) (Figure 6). 

### 3.6. Safety

During the course of treatment, there was one case each of cerebral infarction, deep vein thrombosis, pyoderma gangrenosum, erythema nodosum, and pneumonia, but none of them was determined to be related to VDZ.

## 4. Discussion

This is a multicenter study from Japan showing the efficacy of VDZ even in bio-failure cases, and noting the possibility that an early clinical response may predict long-term prognosis. VDZ has been used in more than 60 countries worldwide since 2014 and was approved as a treatment for UC in Japan in 2018. Since then, data on its efficacy has been reported in Japan [11]. However, the proper use of biologics has not been stipulated. It is important to clarify treatment predictors and factors regarding patients that would indicate a favorable use of VDZ to determine the positioning of VDZ in UC treatment. Reports have shown factors favoring the use of VDZ treatment [28]. Among patient-related factors, age and sex were not associated with a response to VDZ [29,30]. When evaluating disease-related factors, there did not appear to be a relationship between disease duration and efficacy of VDZ [31]. Similarly, no such relationship was shown in our study. An association between low CRP levels and a superior response to VDZ was reported [32]. Head-to-head trials have compared the efficacy of VDZ with other biologics. The VARSITY study was a phase 4 clinical trial that compared VDZ and ADA for treatment of bio naïve UC patients, with results showing the superiority of VDZ for achieving a 1-year CR, endoscopic improvement, and histological improvement [33]. In a network meta-analysis comparing multiple drugs in UC patients, VDZ was reported to be the second most effective agent after IFX in bio-naïve patients and was comparable to ADA in patients with a history of anti-TNF-α antibody therapy [34,35]. VDZ is considered to have relatively low immunogenicity and a low incidence of anti-pharmaceutical antibodies. Naganuma et al. showed that the use of concomitant immunomodulators may be beneficial to maintain the clinical efficacy of VDZ [36]. In our study, both CAI and MES were higher in cases who discontinued treatment with VDZ than in those achieving a long-term CR, and most of those who dropped out had the pancolitis type. On the other hand, CRP levels were not associated with treatment efficacy, and AZA use did not change the long-term efficacy of VDZ. 

Previous reports indicated that the use of VDZ for UC was more effective in bio-naïve cases than in bio-failure cases [8,11,37,38]. In the GEMINI-1 study, the clinical response rate in patients with a history of anti-TNF-α antibody treatment failure was only 38.9%, which tends to be lower than the improvement rate of 53.1% at week 6 in bio-naïve patients [8]. The GEMINI-1 study showed that not only the CR rate, but the sustained improvement rate and mucosal healing rate at week 52 were better in patients with no prior treatment with anti-TNF-α antibody preparations [38]. Also, Motoya et al. reported that the 10-week clinical response rate in bio-naïve patients was 53.2% and 27.1% in bio-failure patients, showing a high therapeutic effect in bio-naïve cases, similar to GEMINI-1 [11]. On the other hand, opposite results have been obtained in some studies. For example, the response to VDZ was independent of previous anti-TNF-α failure in IBD patients [32,39,40,41]. Our study demonstrated that treatment with VDZ was effective in many UC patients in Japan, showing a high mucosal healing rate even in cases with prior biological exposures. Up to 51.7% of the bio-failure patients treated with VDZ as a remission inducer achieved a CR at week 52, and no significant difference was observed in the CR rate between the bio-naïve group and the bio-failure group. Furthermore, more than 80% of our study patients achieved an endoscopic remission 52 weeks after the administration of VDZ. There was no difference in the endoscopic remission rate between bio-naïve and bio-failure groups. Perhaps these results are due to the high AZA utilization in the bio-failure group (48.3% in the bio-failure group, 25.6% in the bio-naïve group). On the other hand, the CR rate was significantly higher in the second bio group than in the third bio group at 6 and 52 weeks after the administration of VDZ. This suggests that VDZ can be a treatment option even in bio-failure cases, but that efficacy may diminish as the number of biotherapeutic drugs used increases. Additionally, we showed that treatment efficacy at week 6 may predict the prognosis of later treatment. Among patients continuing treatment 52 weeks after VDZ administration, most cases had already achieved a CR at week 6 in our study. However, many cases that did not reach a CR at week 6 subsequently dropped out. Therefore, those who did not achieve a CR at 6 weeks were considered to be at high risk of recurrence thereafter. In the GEMINI-1 study, the 52-week mucosal healing rate reached 81.9% in cases who had achieved a clinical response at week 6, and the pathological findings also improved [42,43].

Other institutions in Japan have also reported that early treatment effects are important in predicting a long-term prognosis for VDZ treatment. Nagahori et al. reported that early symptomatic improvement predicted a treatment response at week 10 in TNF- naïve patients [44]. Saito et al. also showed that an early clinical response to VDZ (week 6) can be a favorable predictor of treatment with VDZ for UC [45]. In our study, among 11 cases who had not achieved a CR at week 6, a remission was still possible at week 52 by continued use of VDZ. Four cases had to change treatment due to recurrence, even though they had achieved a CR at 6 weeks. Although results of an evaluation at week 6 alone does not directly affect the subsequent prognosis, these results will be helpful in patient care. It was considered that an early CR can be a predictor of a long-term remission, and week 6 can be an evaluation point for treatment decisions when using VDZ for UC. 

Our study had several limitations. It was a retrospective study with a limited number of cases with different treatment histories. For example, patients in our study had higher rates of usage of corticosteroids or immunomodulators (41.4% and 37.9%, respectively) compared with other studies. Furthermore, it was only possible to evaluate endoscopic severity at week 52 in 30 cases. There also might be a selection bias and inter-observer bias in this study, since decisions on starting VDZ treatment or endoscopic evaluations were made by different physicians in each institution. In addition, we could not examine the correlation between biomarkers, such as LRG or fecal calprotectin, with the therapeutic effect, since methods for using those markers differed among the participating facilities.

## 5. Conclusions

In conclusion, this multicenter study in Japan demonstrated the potential value of VDZ not only in bio-naïve cases but also in bio-failure cases, and showed that early CR may predict long-term prognosis. It is expected that the types of biologics will continue to increase in the future. While treatment options for patients expand, further accumulation of real-world data is required regarding the proper use of each drug so that personalized medicine can be realized.

## Figures and Tables

**Figure 1 biomedicines-10-02526-f001:**
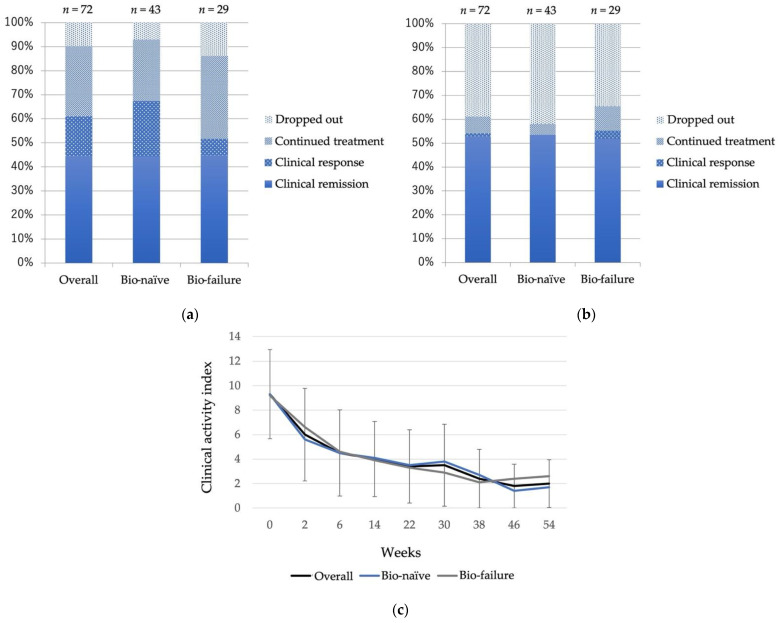
Clinical outcomes of patients treated with Vedolizumab (**a**) at week 6 and (**b**) at week 52. (**c**) Changes of clinical activity index. Treatment efficacy was assessed using the Lichtiger clinical activity index.

**Figure 2 biomedicines-10-02526-f002:**
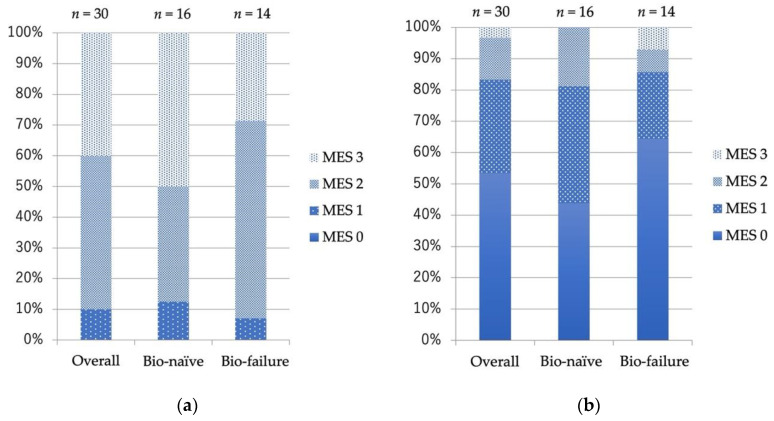
Endoscopic evaluation of patients treated with Vedolizumab at (**a**) baseline, and (**b**) at 52 weeks after Vedolizumab administration. Endoscopic severity was assessed using the Mayo endoscopic score. MES: Mayo endoscopic subscore.

**Figure 3 biomedicines-10-02526-f003:**
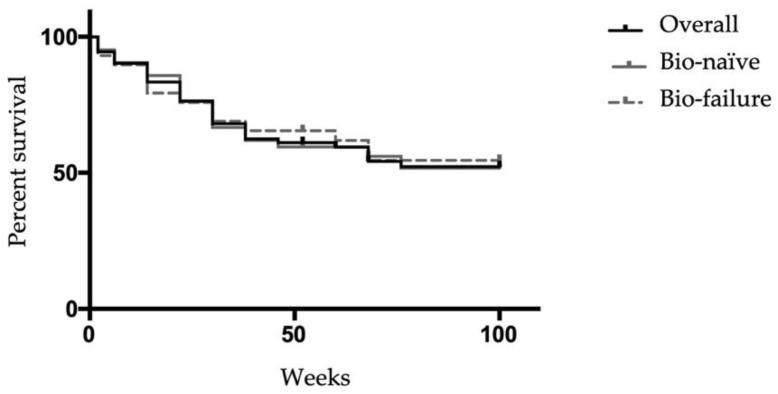
Kaplan–Meier plots for relapse-free survival of patients treated with Vedolizumab up to 100 weeks after administration. (Overall *n* = 51, bio-naïve *n* = 25, bio-failure *n* = 26). Relapse-free survival was assessed using the Kaplan–Meier method.

**Figure 4 biomedicines-10-02526-f004:**
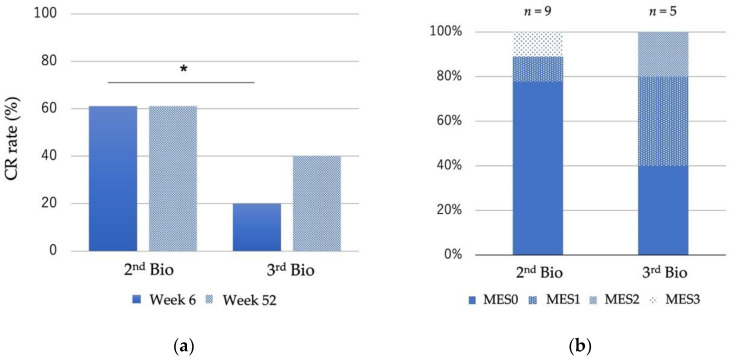
Clinical and endoscopic outcomes of patients treated with Vedolizumab as second bio and third bio. (**a**) Clinical remission rate at week 6 and week 52 (second bio *n* = 18, third bio *n* = 10). Treatment efficacy was assessed using the Lichtiger clinical activity index. (**b**) Endoscopic evaluation at week 52. Endoscopic severity was assessed using the Mayo endoscopic score. Differences between groups were analyzed using the Mann–Whitney’s U test. * *p* < 0.05.

**Figure 5 biomedicines-10-02526-f005:**
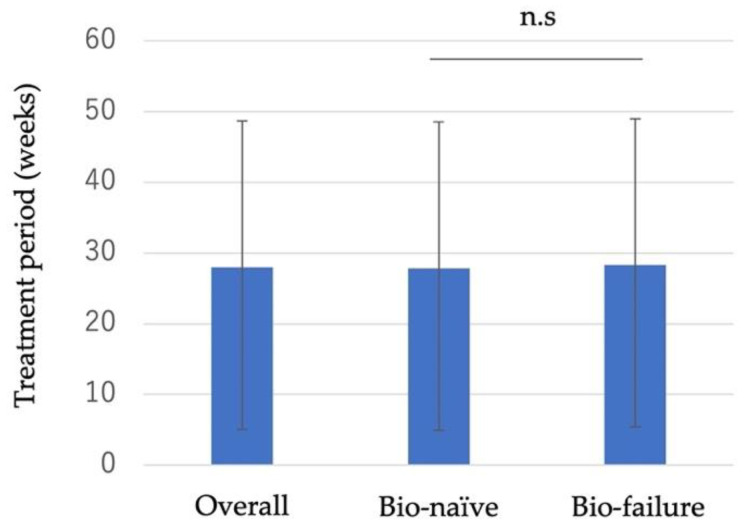
Treatment period in cases where Vedolizumab was discontinued (overall *n* = 33, bio-naïve *n* = 13, bio-failure *n* = 20). Differences between groups were analyzed using Mann–Whitney’s U test. n.s: no significant difference (*p* > 0.05).

**Figure 6 biomedicines-10-02526-f006:**
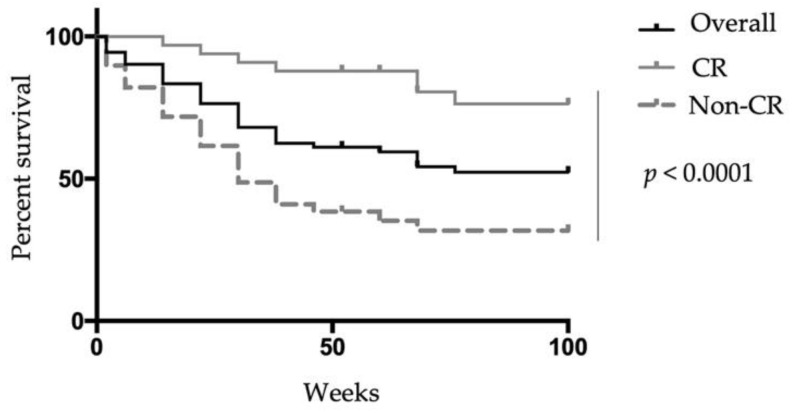
Clinical remission rate and relapse-free survival of patients treated with Vedolizumab. Kaplan–Meier plots for relapse-free survival of patients achieving or unable to achieve a clinical remission at week 6. Relapse-free survival was assessed using the Kaplan–Meier method. CR: Clinical remission.

**Table 1 biomedicines-10-02526-t001:** Patient characteristics.

		(*n* = 87)	
Age, Mean ± SD (range)		45.2 ± 14.0 (20–86)	
Sex, male: female (*n*)		47:40	
Disease duration in years, mean ± SD (range)		9.4 ± 7.5 (1–33)	
Remission induction: maintenance (*n*)		72:15	
Clinical Activity, mean ± SD (range)	Lichtiger CAI ^1^	7.7 ± 4.6 (0–17)	
Location of colitis, *n* (%)	Pancolitis	56 (64.4%)	
	Left-sided colitis	31 (35.6%)	
Mayo endoscopic subscore, *n* (%)	0	13 (14.9%)	
	1	11 (12.6%)	
	2	33 (41.3%)	
	3	30 (37.5%)	
Baseline data, mean ± SD	C-reactive protein (mg/dL)	1.5 ± 1.2	
	Hemoglobin (g/dL)	12.3 ± 2.3	
Bio-naïve, *n* (%)		53 (60.9%)	
Bio-failure, *n* (%): Prior bio use	34 (39.1%)	1	22 (25.3%)
		2	11 (12.6%)
		3	1 (1.2%)
Concomitant drug, *n* (%)	5-Aminosalicylate	78 (89.7%)	
	Prednisolone	36 (41.4%)	
	Azathioprine	33 (37.9%)	

^1^ CAI: clinical activity index.

## Data Availability

The data presented in this study are available on request from the corresponding author.

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
