# Peer review of "Early Clinical Remission Is a Predictor of Long-Term Remission with the Use of Vedolizumab for Ulcerative Colitis"

_biomedicines, 2022, doi:10.3390/biomedicines10102526_

Round 1

Reviewer 1 Report

In this manuscript Keiichi and colleagues present a retrospective, multicentre study on the therapeutic value of Vedolizumab (VDZ) in Inflammatory bowel disease (IBD) and the value of early clinical remission as predictor for long term remission. Using compiled data of 87 patients with moderate to severe Ulcerative Colitis (UC) they show therapeutic value of VDZ in patients with and without previous biological treatment and state that early clinical remission (CR) may predict long-term prognosis. Overall, this is an interesting study. However, I have some major concerns as outlined below:

Major issues

1)        The labelling of the Figures is confusing. I could not find Figure 1, but there are two Figure 3. I assume that Figure 2 is Figure 1, and that the first Figure 3 is indeed Figure 2.

2)        Table 1: The Mayo score of the 24 patients who do not have a score of two or three remains unclear and should be mentioned. The concomitant medication for ulcerative colitis is lacking detailed information about the quantity and quality of this medication. Both should be provided and contain information about the number of patients taking a combination of 5-aminosalicylate, thiopurine, or glucocorticoids and information about the glucocorticoid equivalent dose.

3)        Figure 1 (labelled as Figure 2): I would suggest to also show the change from baseline in clinical activity index in all patients and with and without previous treatment instead.

4)        Figure 2 (labelled as Figure 3): Here the authors present information on the mayo endoscopic score after 52 weeks. However, information on the change from baseline in mayo endoscopic score would add information crucial for the interpretation of the dataset. Furthermore, it seems that only 30 of the 87 patients did receive an endoscopy. I do understand, that not every patient does get an endoscopy after 52 weeks. However, the authors should provide information indicating that these 30 patients are representative for the cohort.

5)        Figure 3 (labelled as Figure 3): Kaplan Meier plots of 51 patients is shown. The author should include information also in the figure legend how these patients were selected out of the 87

5)        Figure 4 a and b: For a better understanding of this data, the authors should provide information on whether the patients from week 6 and week 52 are the same or if only the patients who achieved CR after 6 weeks are shown.

Minor issues

1)             The numbering of the tables and figures is in part inconsistent and confusing.

2)             The font in the graphs is different to the font of the texts.

3)             Please include standard deviations/ interquartile ranges in the graphical presentation of the data.

4)              Figure 6a does not provide additional information and I would consider removing it

Reviewer 2 Report

Comment to the Editors

This retrospective Japanese study investigated the role of early remission as predictor of long-term remission in patients with ulcerative colitis (UC) treated with Vedolizumab (VDZ).

Authors found that early remission was a significant predictor of long-term remission in these patients, and stated that it can be an helpful evaluation point for the treatment decision in UC patients treated with VDZ.

The manuscript is interesting, but some points needs to be addressed.

1. Please add which were the biologics previously used in the not-bionaïve population.

2. How many patients needed dose adjustment to maintain the remission?

3. How did the authors choose the new treatment after VDZ failure?

4. Authors found that there were no difference in obtaining and maintaining remission between bionaïve and not-bionaïve populations. As correctly reported, literature reports conflicting results about this specific point: how the authors explain their results?

Round 2

Reviewer 1 Report

The authors have successfully adressed all my comments and concerns.

Reviewer 2 Report

Authors successfully replied to all questions raised.